REGISTERED REPORT PROTOCOL

# Registered report protocol: A scoping review to identify potential predictors as features for developing automated estimation of the probability of being frail in secondary care

Dirk H. van Dalen[1,2], Angèle P. M. Kerckhoffs[3,4], Esther de Vries[1,2]*

1 Jeroen Bosch Academy Research, Jeroen Bosch Hospital, 's-Hertogenbosch, The Netherlands, 2 Tranzo, Tilburg School of Social and Behavioral Sciences, Tilburg University, Tilburg, The Netherlands, 3 Department of Geriatric Medicine, Jeroen Bosch Hospital, 's-Hertogenbosch, The Netherlands, 4 Department of Nephrology, Jeroen Bosch Hospital, 's-Hertogenbosch, The Netherlands

* e.d.vries@jbz.nl, e.devries@tilburguniversity.edu

## Abstract

### Introduction

The impact of frailty surges, as the prevalence increases with age and the population age is rising. Frailty is associated with adverse health outcomes and increased healthcare costs. Many validated instruments to detect frailty have been developed. Using these in clinical practice takes time. Automated estimation of the probability of being frail using routinely collected data from hospital electronic health records (EHRs) would circumvent that. We aim to identify potential predictors that could be used as features for modeling algorithms on the basis of routine hospital EHR data to incorporate in an automated tool for estimating the probability of being frail.

### Methods

PubMed (MEDLINE), CINAHL Plus, Embase, and Web of Science will be searched. The studied population consists of older people (≥65 years). The first step is searching articles published ≥2018. Second, we add two published literature reviews (and the articles included therein) [Bery 2020; Bouillon, 2013] to our search results. In these reviews, articles on potential predictor variables in frailty screening tools were included from inception until March 2018. The goal is to identify and extract all potential predictors of being frail. Domain experts will be consulted to evaluate the results.

### Discussion

The results of the intended study will increase the quality of the developed algorithms to be used for automated estimation of the probability of being frail in secondary care. This is a promising perspective, being less labor-intensive compared to screening each individual patient by hand. Also, such an automated tool may raise awareness of frailty, especially in those patients who would not be screened for frailty by hand because they seem robust.

**Data Availability Statement:** All relevant data from this study will be made available upon study completion.

**Funding:** The author(s) received no specific funding for this work.

**Competing interests:** The authors have declared that no competing interests exist.

## Conclusion

The identified potential predictors of being frail can be used as evidence-based input for machine learning based automated estimation of the probability of being frail using routine EHR data in the near future.

## Introduction

The concept of frailty can be defined as a state of vulnerability to poor resolution of homeostasis after a stressor event and is a consequence of cumulative decline in many physiological systems during a lifetime [1]. This cumulative decline depletes homeostatic reserves until minor stressor events trigger disproportionate changes in health status [1]. More practical definitions use specific criteria to define frailty as a clinical syndrome (e.g., Fried 2001 [2]). A large systematic review described an overall weighted prevalence of frailty in community dwelling people aged ≥65 years of 10.7% (95% CI = 10.5–10.9%; 21 studies; n = 61,500) [3]. In a study in secondary care patients aged ≥65 years, the frailty rate was 13.9% [4]. Frailty is associated with various adverse health outcomes [5] and increased healthcare costs [6], with the worst outcomes in the frailest [1]. The global impact of frailty is expected to surge, as the prevalence increases with age and the population age is rising [6].

Age is however not the only variable positively associated with frailty. A lot of research work has been performed on predictors of frailty [7]. Many instruments to detect frailty have been developed [7, 8], including a few automated tools to screen for frailty in electronic health records (EHRs) [9–11]. There is no gold standard frailty screening tool for use in clinical practice [6]. Moreover, most frailty measures are based on surveys or questionnaires and are labor-intensive to complete. The widespread use of EHRs enables *automated* estimation of the probability of being frail using routinely collected data, possibly even without using frailty screening instruments. Also, most research on frailty assessment and detection has been performed in community care [12] and is primarily based on specific tools and questionnaires, not on routinely collected health care data [8]. It remains unclear which variables using the routinely collected data from the hospital EHR in automated estimation of the probability of being frail will work best.

Therefore, the aim of our study is to perform a scoping review to build the foundation for the development of an automated tool for estimating the probability of being frail on the basis of routine health care data present in the EHR in secondary care, by identifying all potentially relevant features to test as potential predictors in a modeling effort based on machine learning. To identify existing reviews answering a similar question, the Cochrane database and PubMed were searched. In the Cochrane database, none were found. In PubMed, two consecutive review articles answering a similar question were identified [8, 13]. These reviews, however, extracted variables from frailty instruments, where our aim is to identify all potentially relevant predictors for frailty that can be extracted from the hospital EHR, not limited to frailty instruments only.

Therefore, we will perform an extensive search in a broad corpus of published literature since 2018. The earlier mentioned reviews (and the articles included therein) will then be added to our search results [8, 13]. We strive to identify all potentially relevant predictors for the presence of frailty that can be extracted from the hospital EHR, not limited to frailty instruments only. The retrieved possible predictors will be shared and discussed with experts in the

geriatric domain to evaluate their potential value in everyday clinical practice. This paper contains the protocol for our literature search and subsequent domain expert evaluation.

## Methods

This protocol is written according to the Preferred Reporting Items for Systematic review and Meta-Analysis protocols (PRISMA-P) guidelines (See S1 Appendix).

### Literature search

Four databases will be searched for eligible articles: PubMed (MEDLINE), CINAHL Plus, Embase, and Web of Science. Eligible articles are written in English or Dutch. Articles that we were not able to get hold of as full-text will be reported as not retrieved. Case studies will be excluded, because we consider case studies not suitable for providing information on potential predictor variables. Review articles will be included but, as reviews are secondary research, only in order to retrieve the reviewed original articles in case we did not find them. The studied population consists of older people ($\geq$65 years). The key inclusion criterion is information on possible predictors for frailty. This is not restricted to any specific type or measure of frailty. Articles only describing frailty as an independent variable to predict another outcome (e.g., mortality), and not including frailty itself as an intermediate or outcome variable will be excluded.

The search strategy is based on Medical SubHeadings (MeSH) in PubMed (MEDLINE), Medical Headings (MH) in CINAHL Plus, and the Emtree thesaurus in Embase. Web of Science does not have such a thesaurus or list of subject terms. Next, key terms and synonyms retrieved from an explorative search are searched in titles and abstracts. The search string contains four building blocks which are combined using Boolean operators ('OR' within the building blocks, 'AND' between the building blocks). The first building block is designed to find articles about the main topic: frailty. It is a combination of MeSH and title/abstract terms, in order to find all research articles on frailty. The second building block is defined to retrieve research articles with a method appropriate for assessing predictors for frailty. The third building block contains terms and synonyms to search articles within the intended study population. The fourth building block is narrowing down the scope to only those articles literally describing possible predicting factors in title or abstract. Therefore, this building block only contains title/abstract terms. PubMed (MEDLINE), CINAHL Plus, Embase, and Web of Science, will be searched for articles published $\geq$2018 until the present time. Filters are used for Dutch or English language and age $\geq$65 years. As the filter on age uses the index, and indexing is delayed, the search string will be split in two separate parts: articles published from 2021 onwards will not be filtered on age. The complete search string for PubMed (MEDLINE) can be found in Box 1. The complete PubMed (MEDLINE) search strategy can be found in S2 Appendix. CINAHL Plus, Embase, and Web of Science will be searched using the same strategy with the search string adapted to the CINAHL Plus, Embase, and Web of Science database, respectively. The CINAHL Plus, Embase, and Web of Science search strings can be found in S3 Appendix. The review articles of Bery (2020) and Bouillon (2013) (and the articles included therein) will be added to our search results [8, 13]. The goal is to identify and extract as many potential predictors of the presence of frailty as possible.

### Study selection

The study selection takes place according to the eligibility criteria. First, title selection will be performed by at least two authors independently. In case of discordance, the study will be included in the abstract selection. Second, abstract selection will be performed by at least two

## Box 1. Search string PubMed (MEDLINE)

(((("Frail Elderly"[Mesh] **OR** "Frailty"[Mesh] **OR** "Functional status"[Mesh] **OR** "Frail*"[tiab] **OR** "Debilit*"[tiab] **OR** "geriatric syndrome*"[tiab] **OR** "Pre-frail*"[tiab] **OR** "Functional status" [tiab] **OR** "Fragil*"[tiab] **OR** "Vulnerab*"[tiab] **OR** "Resilien*"[tiab]) **AND** ("Risk Assessment"[Mesh] **OR** "Surveys and Questionnaires"[Mesh] **OR** "tool*"[tiab] **OR** "instrument*"[tiab] **OR** "predictive model*"[tiab] **OR** "prediction model*"[tiab] **OR** "questionnaire*"[tiab] **OR** "Risk assessment" [tiab] **OR** "index*"[tiab] **OR** "inventor*"[tiab] **OR** "survey*"[tiab] **OR** "assessment method*"[tiab]) **AND** ("Aged"[Mesh] **OR** "Geriatrics"[Mesh] **OR** "Aged"[tiab] **OR** "Elde*"[tiab] **OR** "Olde*"[tiab] **OR** "geriatric*"[tiab] **OR** "centenarian*"[tiab] **OR** "centarian*"[tiab] **OR** "nonagenarian*"[tiab] **OR** "octogenarian*"[tiab] **OR** "octagenarian*"[tiab] **OR** "septuagenarian*"[tiab] **OR** "very old"[tiab] **OR** "senior*"[tiab]) **AND** ("risk factor*"[tiab] **OR** "variable*"[tiab] **OR** "predictor*"[tiab] **OR** "parameter*"[tiab] **OR** "deficit*"[tiab] **OR** "characteristic*"[tiab] **OR** "determinant*"[tiab] **OR** "criteri*"[tiab]) **AND** (dutch[la] **OR** english[la]) **AND** (aged[filter]) **AND** (2018:2020[pdat])) **OR** (("Frail Elderly"[Mesh] **OR** "Frailty"[Mesh] **OR** "Functional status"[Mesh] **OR** "Frail*"[tiab] **OR** "Debilit*"[tiab] **OR** "geriatric syndrome*"[tiab] **OR** "Pre-frail*"[tiab] **OR** "Functional status" [tiab] **OR** "Fragil*"[tiab] **OR** "Vulnerab*"[tiab] **OR** "Resilien*"[tiab]) **AND** ("Risk Assessment"[Mesh] **OR** "Surveys and Questionnaires"[Mesh] **OR** "tool*"[tiab] **OR** "instrument*"[tiab] **OR** "predictive model*"[tiab] **OR** "prediction model*"[tiab] **OR** "questionnaire*"[tiab] **OR** "Risk assessment" [tiab] **OR** "index*"[tiab] **OR** "inventor*"[tiab] **OR** "survey*"[tiab] **OR** "assessment method*"[tiab]) **AND** ("Aged"[Mesh] **OR** "Geriatrics"[Mesh] **OR** "Aged"[tiab] **OR** "Elde*"[tiab] **OR** "Olde*"[tiab] **OR** "geriatric*"[tiab] **OR** "centenarian*"[tiab] **OR** "centarian*"[tiab] **OR** "nonagenarian*"[tiab] **OR** "octogenarian*"[tiab] **OR** "octagenarian*"[tiab] **OR** "septuagenarian*"[tiab] **OR** "very old"[tiab] **OR** "senior*"[tiab]) **AND** ("risk factor*"[tiab] **OR** "variable*"[tiab] **OR** "predictor*"[tiab] **OR** "parameter*"[tiab] **OR** "deficit*"[tiab] **OR** "characteristic*"[tiab] **OR** "determinant*"[tiab] **OR** "criteri*"[tiab]) **AND** (dutch[la] **OR** english[la]) **AND** (2021:2022[pdat]))))

This box displays the search string for PubMed (MEDLINE) in step 1, including filters on age, language, and year of publication. Abbreviations: Mesh = Medical subheading; tiab = title / abstract; la = language; pdat = publication date.

authors independently. In case of discordance, the study will be included in the full text selection. The last step in the selection process is selection of full texts, again by at least two authors independently. In case of discordance, a third author will be consulted in a group discussion until consensus is reached. The selection process will be performed using Rayyan, a web and mobile app for systematic reviews [14]. The selection process is shown in Fig 1.

### Data collection

The full texts of all selected articles will be scanned by at least two authors independently in order to extract all potential predictors for frailty, starting with the latest publication. In case of discordance, a third author will be consulted in a group discussion until consensus is reached. In this study, a variable is a single data point (e.g. question, item, clinical value, or test result) and is considered a potential predictor when it is described as a factor possibly related to frailty, irrespective of the variable type and its described significance in that article.

### Data extraction

The metadata of the articles (i.e., author(s), year of publication, study name, study design, study setting, and study country), descriptive baseline variables (i.e., study population, age, sex, and number of subjects), and all unique potential predictors of being frail (including relevant information such as the potential predictor type and the definition of frailty used in the article where a potential predictor was mentioned; Table 1) will be extracted and recorded in Microsoft Excel. No risk of bias assessment will be performed because no effects on endpoints are quantified. Also, irrelevant features will automatically be dropped in the future machine

| Title selection | 1. Is the study considered relevant based on the title? | Yes: Include title for abstract selection.<br>Maybe: Include title for abstract selection.<br>No: Exclude title (code E1). |
|---|---|---|
| Abstract selection | 2. Is the study considered relevant based on the abstract? | Yes: Include abstract for full text selection.<br>Maybe: Include abstract for full text selection.<br>No: Exclude (code E2). |
| Full text selection | 3. Is the full text available in English or Dutch language? | Yes: Proceed to 4.<br>No: Exclude (code E3). |
| | 4. Is the study a case study? | No: Proceed to 5.<br>Yes: Exclude (code E4). |
| | 5. Does the study population include patients aged ≥65 years? | Yes: Proceed to 6.<br>No: Exclude (code E5). |
| | 6. Is frailty described as dependent or intermediate variable (and not as independent variable only? | Yes: Proceed to 7.<br>No: Exclude (code E6). |
| | 7. Are independent variables for frailty described? | Yes: Proceed to 8.<br>No: Exclude (code E7). |
| | 8. Is the study a review study? | No: Include (code I1).<br>Yes: Include(code I2). |

If an article is not included based on the above criteria but is considered relevant as background, add code B (e.g. E6B).

**Fig 1. Selection process.** The figure shows the selection process of the retrieved records.

learning process after this review has been completed. In that machine learning process we aim to use different methods of supervised (e.g. regression and random forest) and unsupervised (e.g. classification and clustering) learning. In the supervised machine learning methods, frailty will be defined as frailty present in steady state measured by the available reported results of frailty screening tools in the patient's EHR. Finally, we intend to compare the outcome of the developed algorithm with frailty measured by comprehensive geriatric assessment

**Table 1. Format for collecting extracted potential predictors for frailty.**

| Potential predictor |
|---|
| Potential predictor type |
| Total count of articles in which potential predictor is positively correlated with frailty |
| Total count of articles in which potential predictor is not correlated with frailty |
| Total count of articles in which potential predictor is negatively correlated with frailty |
| Definition of frailty used in the study where the potential predictor was mentioned |
| Free text notes |
| Relevant information about potential predictor (to be determined based on retrieved information) |

(CGA) (if available in the patient's EHR) and/or consensus among (inter)national domain experts upon reviewing the patient's EHR as 'gold standard'.

## Synthesis of results

The results will be described in an extensive categorized table of all unique potential predictors for being frail, including total count of articles in which a potential predictor was mentioned and other relevant information such as the potential predictor type and the definition of frailty used in the article where a potential predictor was mentioned. The count of how many times a variable is mentioned and if an association with frailty is present or absent will not be used to summarize evidence of an association or to draw any conclusions about associations. Potential predictors will be categorized using the same (sub)headings as used in the layout of most EHRs (such as medical history, physical examination, medications, laboratory results, radiological imaging, etc.).

## Collection of domain expert evidence

After retrieving and listing all potential predictors for frailty, three domain experts working in different hospital settings (secondary, tertiary, and international, respectively) will be asked whether they miss relevant articles and/or potential predictors in the list of included articles and the database with collected potential predictors. The expert feedback (additional articles and/or potential predictors) will be added to the results. If the experts judge a potential predictor in our list as not relevant, this will be noted in the list, but the potential predictor will not be deleted. If the experts mention a relevant article which was not included in our study, it will be checked for additional potential predictors. Retrieved additional potential predictors will be added to the list, including a note that this predictor was introduced by the experts.

## Data management

Data will initially be stored in the secure digital environment of the Jeroen Bosch Hospital. After completion of this study, all data regarding the study will be stored at Tilburg University in a secured environment at the department Tranzo. Data will be retained for 15 years. After publication of the results, the data will be available on request.

## Timeline

The study will start as soon as the study protocol is accepted for publication in the Registered Reports section of PLOS ONE. The study report is planned to be completed and ready for submission in the year following the acceptance of the protocol.

## Current status

An exploratory search in PubMed (MEDLINE), CINAHL Plus, Embase, and Web of Science resulted in a total of 32.526 records (7.796 PubMed (MEDLINE), 2.722 CINAHL Plus, 6.147 Embase, and 15.861 Web of Science, these were not yet checked for duplicate records) (Fig 2).

## Discussion

The described study is designed to identify as many potentially relevant variables as possible, to include as potential predictors for the estimation of the probability of being frail in a modeling effort using machine learning. The retrieved potential predictors will be used to build the foundation for the development of algorithms to be used for automated estimation of the

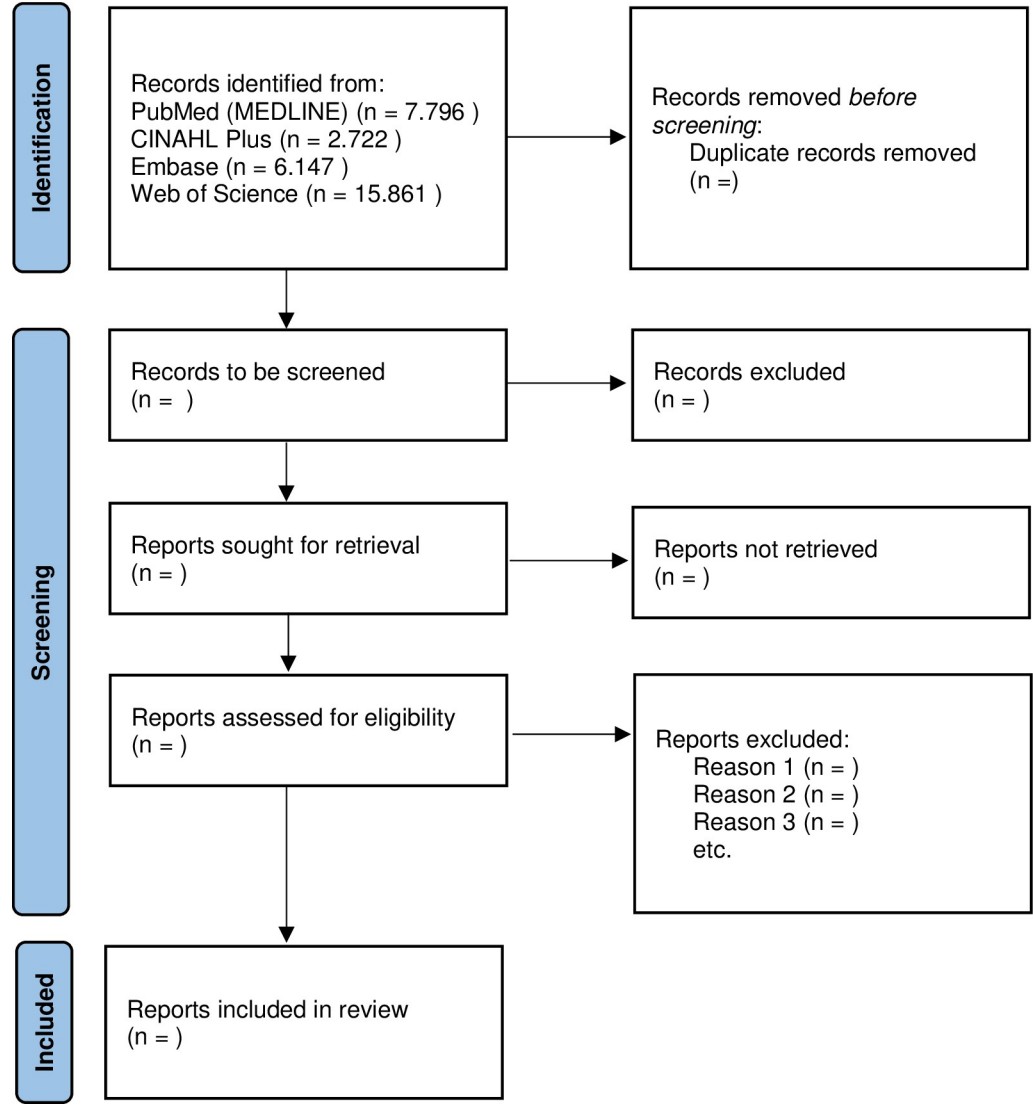

**Fig 2. PRISMA flow diagram.** The flow diagram shows the identification, screening, and inclusion of the identified records through PubMed (MEDLINE), CINAHL Plus, Embase, and Web of Science.

probability of being frail on the basis of routine health care data collected from the EHR in secondary care.

This effort is important because the prevalence of frailty is increasing due to ageing [6] while, as a result of developments in medicine, treatment options are also increasing. On the other hand, frailty is associated with an increased risk of postoperative complications [15]. Therefore, it is useful to predict the risk of being frail in order to assess eligibility for treatment [6]), or to identify those who might benefit from an intervention to reduce frailty [16].

Many frailty measures have been developed, but most are based on surveys or questionnaires and are labor-intensive to complete. The widespread use of EHRs enables automated estimation of the probability of being frail using routine care data, possibly even without using frailty screening instruments. Some tools using EHR data to predict frailty and with the ability to be fully automated have been proposed, however most of them use unweighted frailty indices like the electronic Frailty Index (eFI), and all with a slightly different set of deficits [9–11,

17–19]. Thus, it is not yet clear which included variables using routinely collected data from the hospital EHR will work best. This study provides an overview of all potential predictors of frailty and adds advancements for automated estimation of the probability of being frail in secondary care. This is a promising perspective, being much less labor-intensive compared to screening each individual patient by hand. Also, it may raise awareness of frailty, especially in patients who are not screened for frailty because they seem robust.

The study has several strengths. First, it is pre-registered, which enables peer-review prior to the start of the study as well as repetition of the study. Second, the data collection and extraction process is carried out by multiple researchers, reducing the risk of bias. Third, the information retrieved in the study will be evaluated by experts in the domain of frailty for completeness. The study also has limitations. The study is designed to collect potential predictors in the most recent published literature. As we use two earlier published reviews as a starting point and the results are cross-checked with domain expert evidence, we are confident that our literature search will encapsulate the most relevant potential predictors for frailty.

## Conclusion

The extensive list of potential predictors of frailty provided by this study can be used as an evidence-based foundation for a modeling effort using machine learning to develop algorithms to be used for automated estimation of the probability of being frail based on hospital EHR variables recorded in routine care in the near future.

## Supporting information

**S1 Appendix. Preferred reporting items for systematic review and meta-analysis protocols (PRISMA-P) checklist.**
(DOCX)

**S2 Appendix. PubMed (Medline) search.**
(DOCX)

**S3 Appendix. CINAHL Plus, Embase, and Web of Science search.**
(DOCX)

## Acknowledgments

We would like to sincerely thank prof. dr. Marcel A.L.M. van Assen (Tilburg University and Utrecht University) for his contribution in improving the final manuscript.

## Author Contributions

**Conceptualization:** Angèle P. M. Kerckhoffs, Esther de Vries.

**Data curation:** Dirk H. van Dalen.

**Methodology:** Dirk H. van Dalen, Angèle P. M. Kerckhoffs, Esther de Vries.

**Supervision:** Angèle P. M. Kerckhoffs, Esther de Vries.

**Writing – original draft:** Dirk H. van Dalen.

**Writing – review & editing:** Dirk H. van Dalen, Angèle P. M. Kerckhoffs, Esther de Vries.

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
