## [Decision Letter · Decision Letter 0]

19 Apr 2022

PONE-D-21-38902Identifying potential predictors for automated frailty detection. A registered report protocolPLOS ONE

Dear Dr. de Vries,

Thank you for submitting your manuscript to PLOS ONE. After careful consideration, we feel that it has merit but does not fully meet PLOS ONE’s publication criteria as it currently stands. Therefore, we invite you to submit a revised version of the manuscript that addresses the points raised during the review process. Please see my specific comments at the end of this message.

We look forward to receiving your revised manuscript.

Kind regards,

Edison I.O. Vidal, MD, MPH, PhD

Academic Editor

PLOS ONE

Journal Requirements:

3. Please include a PRISMA-P checklist, for protocols, instead of a PRISMA checklist, as per our submission guidelines for registered report protocols of systematic reviews. For more information, see https://journals.plos.org/plosone/s/submission-guidelines#loc-guidelines-for-specific-study-types.

Additional Editor Comments:

I have evaluated this systematic review protocol and the reviewers’ comments, which I found very helpful.

1. As reviewer #1 pointed out the text is very unclear regarding its aims. On the one hand, the authors argue about the importance of detecting frailty. On the other hand, they describe the aim of their study and adopt eligibility criteria designed to identify predictors of frailty. Detecting frailty is very different from predicting frailty and has major implications for the design of this review. In the first paragraph of the discussion section, the authors make the following statement: “The described study is designed to identify as many potentially relevant variables as possible, to test as predictors for frailty in a modeling effort. The retrieved potential predictors will be used to build the foundation for the development of an automated frailty detection tool on the basis of routine health care data collected from the EHR in secondary care.” If the final aim of the authors is indeed as they wrote in that last sentence, then it makes little sense to look for predictors of frailty. It is essential that the authors attain more clarity regarding the true aims of their study and that their justifications and methods are aligned with those goals.

2. It is insufficient to conduct searches only in Pubmed and CINAHL. At the very least, the authors must also search EMBASE e Web of Science.

3. Lines 96 to 98: “Two databases will be searched for eligible articles: PubMed (MEDLINE) and CINAHL Plus. Eligible articles are written in English or Dutch and should have an available full text to enable complete review.” The availability of a full text is often related to the authors' resources and tenacity. It is inappropriate to exclude a reference because its text was not available online in the databases subscribed by the authors' institution or in their library.

4. The methodological approach to start selecting articles published from 2018 onwards and looking for “data saturation” is not acceptable for a rigorous systematic review.

5. Lines 147 to 150: “First, title selection will be performed by the first author. To reduce the risk of bias, a random sample of 10% of the titles will be blinded and double checked by the second author, and another random sample of 10% of the titles by the third author.” Lines 151 to 153: “If discordance between two authors is ≥20%, a group discussion to fine tune the eligibility criteria will be held, and the first author will then repeat the title selection.” That approach is not acceptable for a high-quality systematic review. At least two authors must screen and select studies independently. The same observation applies to the data extraction process.

6. Line 178 to 180: “In this study, a variable is a single data point (e.g. question, item, clinical value, test result) and is considered a potential predictor when it is described as a factor possibly related to frailty, irrespective of its described significance in that article.” If what the authors aim is only the identification of predictors/risk factors for frailty, then they should also be interested in their statistical significance.

7. Lines 186 to 188: “The extraction and selection will be an iterative process in which searching, selecting, and extracting potential predictors is repeated systematically, to ensure adequate data collection.” It is unclear if such an iterative process makes sense in a review like this one that apparently will not address qualitative data, although this is not clear in the eligibility criteria.

8. Lines 188 to 189: “No risk of bias assessment will be performed because no effects on endpoints are quantified.” Systematic reviews must perform evaluations of risk of bias and of the overall certainty of evidence. The authors’ argument that “no effects on endpoints are quantified” as a justification is flawed. The only reason why endpoints are not quantified is that the authors intend only to “count the votes” of included studies in terms of positive, negative, and absent associations. Vote counting is one of the worst possible strategies to perform a literature synthesis and is not acceptable for any high-quality systematic review.

9. I found reviewer #3 comment on the question of acute care data extremely relevant because it is often important to differentiate frailty present before hospital admission from frailty developed because of an acute problem immediately before or in the course of hospital admission.

10. I also agree with reviewer #3 that the authors should read Searle’s 2008 article, which provides guidance on the development of frailty indexes, which can also be used with EHR.

Reviewers' comments:

Reviewer's Responses to Questions

**Comments to the Author**

1. Does the manuscript provide a valid rationale for the proposed study, with clearly identified and justified research questions?

Reviewer #1: Partly

Reviewer #2: Partly

Reviewer #3: Yes

2. Is the protocol technically sound and planned in a manner that will lead to a meaningful outcome and allow testing the stated hypotheses?

Reviewer #1: Partly

Reviewer #2: Partly

Reviewer #3: Yes

3. Is the methodology feasible and described in sufficient detail to allow the work to be replicable?

Reviewer #1: Yes

Reviewer #2: Yes

Reviewer #3: Yes

4. Have the authors described where all data underlying the findings will be made available when the study is complete?

Reviewer #1: Yes

Reviewer #2: Yes

Reviewer #3: Yes

5. Is the manuscript presented in an intelligible fashion and written in standard English?

Reviewer #1: Yes

Reviewer #2: Yes

Reviewer #3: Yes

6. Review Comments to the Author

You may also provide optional suggestions and comments to authors that they might find helpful in planning their study.

Reviewer #1: This is an interesting protocol of a systematic review, in which the authors aim to identify predictors for frailty that could serve as a foundation for future development of EHR-based frailty measures in secondary care. As there has been increasing efforts in various countries to develop automatic frailty tools in clinical settings, this work is important and relevant to the field. I also think that this protocol is well-written, with clear and transparent descriptions on the literature search process, data management, and timeline. I have a few comments below, which I hope can further help in improving the clarity of the aims and methods of the protocol:

- My main concern is on the meaning of the word “predictor”, which is not entirely clear to me in this context of EHR-based frailty. I think it can mean slightly different things depending on the operational approach to frailty. If I understand correctly, the authors would like to identify both risk factors for frailty and variables for creating the frailty measures (lines 85–87)? For the latter, I am not sure if I would call it a predictor if, for example, the health deficit variables in creating a Rockwood frailty index are embedded as part of the frailty construct. I think the authors can perhaps give some more examples of frailty measures and predictors in the Introduction to make it clearer.

- Similarly, I would also expect some descriptions in the Methods on how frailty will be defined. Do the authors plan to include all available frailty measures? “Frailty” can mean quite different things (and the predictors may be different) for a frailty phenotype, a deficit-based frailty index, a data-driven frailty score, or the Clinical Frailty Scale.

- Another thing I also feel a bit confused is whether the authors are searching only articles for EHR-based frailty measures, or any frailty measures in general that are used in secondary care? I think this can be stated more clearly in the aim.

- The authors emphasized a lot on the potential development of an automated frailty detection tool based on the predictors identify from this study. A bit more information on that will be of interest; for example, are the authors referring to some machine learning methods to build automated frailty tools?

- I don’t think it is accurate to say that “none of them [automated tools using EHR data] in secondary care” (line 248). There are at least several EHR-based frailty indices (PMID: 33770164, 34967685, 34650997), as well as the Hospital Frailty Risk Score (PMID: 29706364) developed for frailty detection in secondary care. Please consider rephrasing it.

- The authors may consider stating explicitly in the title that it is a systematic review (and it is frailty for secondary care) to help the readers understand the study design.

Reviewer #2: The authors are embarking on a worthwhile endeavor (aimed at better using ROUTINE healthcare data for frailty) but a few points need more careful consideration

1) It is unclear exactly how this proposed review differs from the work of Bery et al (ref 8). Yes the authors want to include more than just frailty instruments - but wouldn't most of the articles yielded by the search terms (Box 1) be frailty scores? If not, then the authors should outline what kind of articles they seek to capture (in addition to frailty scores/measures). I worry that not specifying this will yield a set of articles that is too heterogeneous

2) More clarification is needed on the endpoint of the search ("data saturation"). The search outlined will possibly yield many thousand articles. What if each article generates new variables not previously seen (note that the "accumulation of deficits frailty approach" gives a near-infinite number of possible deficits). How will they deal with this volume of articles?

3) As the authors may know, there are two major "schools of thought" regarding frailty measurement. The accumulation of deficits approach VS the phenotype approach. In studies such as the present study, the accumulation of deficits approach will be OVER-represented as these frailty scores can use a wide variety of variables (in fact, any variable is allowed as long as there are enough deficits). The authors should outline whether they will record the approach used for each study their review identifies.

4) Risk of bias - the authors should consider doing a more systematic "risk of bias" assessment (rather than just discrepancy checking amongst the authors). It will be helpful to consider how to grade the quality of articles included in the final review

Reviewer #3: This manuscript summarizes the protocol for a literature review to identify predictors as part of automated frailty detection using the electronic health record. Authors attest that results reported have not been published elsewhere.

The process described will include a systematic review and will consider multiple possible sources of data from the inpatient hospital sphere for frailty.

Major Concerns:

1. While frailty is defined correctly according to the most prominent definitions, the manuscript authors do not specifically lay out how their predictors will be selected beyond that they have been used elsewhere. This is unfortunate, because the concept of frailty detection using the frailty index methodology is well described and - significantly - shows that it is vitally important that the elements selected are related to aging. Strongly recommend that the authors review Searle et al. 2008 BMC Geriatrics for a "recipe" for a frailty index.

2. The authors note that most frailty assessments and indices are in community care (primary care in US). They note two particular frailty indices used in UK (Clegg et al) and US (Pajewski et al). Both use routinely collected data. However, they do not engage with the question of using acute care data (when the patient is not in a "steady state" for frailty assessment. Does this truly reflect frailty, or does it reflect acuity of illness? some consideration of this concern should be integrated into the plan.

3. No description is listed for how authors will determine whether or not to include elements in their frailty assessment, or how closely related a concept is to frailty, or whether to weight items as in traditional risk prediction models or weight all as "1" as in frailty indices

4. The article is clear and is written in standard English.

Given the above serious concerns, I would not recommend this for publication until and unless the authors engage with the extant literature on hospital-based frailty indices, the pros/cons of defining frailty using acute hospital (versus primary care) data, and the existing guidelines for building a frailty index.

7. PLOS authors have the option to publish the peer review history of their article (what does this mean?). If published, this will include your full peer review and any attached files.

Reviewer #1: **Yes: **Jonathan Mak

Reviewer #2: No

Reviewer #3: No

---

## [Author Response · Author response to Decision Letter 0]

24 Jun 2022

Response to reviewers

PONE-D-21-38902

Identifying potential predictors for automated frailty detection. A registered report protocol

PLOS ONE

Dear editor, dear reviewers,

Thank you for the useful comments and suggested revisions. We are pleased to read about the potential value and uniqueness of our paper. Hereby we submit our revision of the manuscript. Our revisions can be found in the tracked changes in the manuscript. Our response to the comments and suggestions made by the editor and reviewers can be found below, in italics. Thank you for your consideration of these responses and revisions.

Response to the editor

We checked the author guidelines and the journal requirements and adapted the manuscript accordingly. Furthermore, we have amended the filenames as requested. Unfortunately, erroneous file naming was probably the reason why the Prisma-P checklist was not recognized as such. We have reviewed its content and amended the file name.

Additional Editor Comments:

Editor: I have evaluated this systematic review protocol and the reviewers’ comments, which I found very helpful.

Editor: 1. As reviewer #1 pointed out the text is very unclear regarding its aims. On the one hand, the authors argue about the importance of detecting frailty. On the other hand, they describe the aim of their study and adopt eligibility criteria designed to identify predictors of frailty. Detecting frailty is very different from predicting frailty and has major implications for the design of this review. In the first paragraph of the discussion section, the authors make the following statement: “The described study is designed to identify as many potentially relevant variables as possible, to test as predictors for frailty in a modeling effort. The retrieved potential predictors will be used to build the foundation for the development of an automated frailty detection tool on the basis of routine health care data collected from the EHR in secondary care.” If the final aim of the authors is indeed as they wrote in that last sentence, then it makes little sense to look for predictors of frailty. It is essential that the authors attain more clarity regarding the true aims of their study and that their justifications and methods are aligned with those goals.

Response: We agree with the reviewer that our aims were not fully clear, and we want to thank the reviewer for raising this important point. First of all, we do not intend to perform a systematic review, but a scoping review. Also, we would like to clarify that we intend to carry out this review to gather input for building an automated tool to predict the risk of someone being frail, not to develop another frailty detection instrument.

Since our aim was not entirely clear, we have clarified the text regarding this issue at several points in the manuscript , most explicitly in line 95-99 of the revised manuscript with track changes: “Therefore, the aim of our study is to perform a scoping review to build the foundation for the development of an automated prediction tool for the risk of frailty on the basis of routine health care data present in the EHR in secondary care, by identifying all potentially relevant features to test as potential predictors in a modeling effort based on machine learning”.

Furthermore, we have replaced all phrases referring to the ‘frailty detection tool’ with ‘tool for prediction of the risk of frailty’ throughout the entire manuscript. As the identified potential predictors actually are predictors for frailty, we left all phrases referring to these as they were. We will use machine learning to develop that automated tool and will include all potential predictors for frailty identified through this review while creating the final algorithms. During the machine learning process, variables will be dropped when they do not increase the predictive capacity of the algorithms. 

We clarified the above throughout our manuscript, and added: “Also, irrelevant features will automatically be dropped in the future machine learning process after this review has been completed.”, which can be found in line 223-224 of the revised manuscript with track changes.

Editor: 2. It is insufficient to conduct searches only in Pubmed and CINAHL. At the very least, the authors must also search EMBASE e Web of Science.

Response: We agree with the reviewer that for a systematic review searching in only PubMed and CINAHL would be insufficient. However, we are not performing a systematic but a scoping review with the aim to identify as many potential predictors as possible for our future machine learning efforts. But we agree with the reviewer that searching Embase would strengthen this review. We have amended the manuscript and included Embase in our search strategy. 

We added this step to our manuscript: “Three databases will be searched for eligible articles: PubMed (MEDLINE) and), CINAHL Plus, and Embase.”, which can explicitly be found in line 116-117 of the revised manuscript with track changes and further throughout the methods section. Furthermore, we added the Embase search strategy to S2 Appendix.

Editor: 3. Lines 96 to 98: “Two databases will be searched for eligible articles: PubMed (MEDLINE) and CINAHL Plus. Eligible articles are written in English or Dutch and should have an available full text to enable complete review.” The availability of a full text is often related to the authors' resources and tenacity. It is inappropriate to exclude a reference because its text was not available online in the databases subscribed by the authors' institution or in their library.

Response: We agree with the reviewer that a reference cannot be excluded just because no full text is available online. We would like to take away the reviewer’s concerns by explaining that we can retrieve practically any reference through the Dutch Royal Library (https://www.kb.nl/en). Furthermore, in the rare case that a full text is not available in the Dutch Royal Library, we will request the corresponding author by e-mail to share the article’s full text. Only if that is still not successful will we exclude the article. 

We did not amend the manuscript on this point.

Editor: 4. The methodological approach to start selecting articles published from 2018 onwards and looking for “data saturation” is not acceptable for a rigorous systematic review.

Response: As stated earlier, we intend to perform a scoping review and not a systematic review. We aim to map the existing literature in an extensive table containing all potential predictors of frailty, irrespective of how many times they were described or whether they were used in a developed frailty index. As we expect to find overlap in the mentioned potential predictors in the literature, we believe our method is able to provide an oversight of all possibly relevant potential predictors described in the literature. Moreover, by not continuing the search after data saturation has been reached, we prevent an enormous amount of work that would not lead to any increase in the yield of our research. 

To describe our interpretation of data saturation more clearly, we added this at two points in the revised manuscript with track changes (line 49 and 322: “(i.e., no new potential predictors are found)”).

Editor: 5. Lines 147 to 150: “First, title selection will be performed by the first author. To reduce the risk of bias, a random sample of 10% of the titles will be blinded and double checked by the second author, and another random sample of 10% of the titles by the third author.” Lines 151 to 153: “If discordance between two authors is ≥20%, a group discussion to fine tune the eligibility criteria will be held, and the first author will then repeat the title selection.” That approach is not acceptable for a high-quality systematic review. At least two authors must screen and select studies independently. The same observation applies to the data extraction process.

Response: We agree with the reviewer that the study selection and data extraction must be performed by at least two authors independently, although we will conduct a scoping review and not a systematic review. 

We amended the manuscript accordingly for title, abstract and full-text selection in line 172-195 of the revised manuscript with track changes: “First, title selection will be performed by at least two authors independently. In case of discordance, the study will be included in the abstract selection. Second, abstract selection will be performed by at least two authors independently. In case of discordance, the study will be included in the full text selection. The last step in the selection process is selection of full texts, again by at least two authors independently. In case of discordance, a third author will be consulted in a group discussion until consensus is reached.”; and for data extraction in line 202-208 of the revised manuscript with track changes: “The full texts of all selected articles will be scanned by at least two authors independently in order to extract all potential predictors for frailty, starting with the latest publication. In case of discordance, a third author will be consulted in a group discussion until consensus is reached.”.

Editor: 6. Line 178 to 180: “In this study, a variable is a single data point (e.g. question, item, clinical value, test result) and is considered a potential predictor when it is described as a factor possibly related to frailty, irrespective of its described significance in that article.” If what the authors aim is only the identification of predictors/risk factors for frailty, then they should also be interested in their statistical significance.

Response: We understand the point made by the reviewer; this would be essential in case we wanted to perform a systematic review. However, as mentioned above our intention is to perform a scoping review. Our aim is to identify all potential predictors from the literature, irrespective of their earlier described statistical significance, and use machine learning to perform feature selection based on a dataset containing the EHR data of an expected 15,000 – 20,000 patients. Keeping in mind that also small sample studies with a possible risk of being underpowered will be considered for inclusion in this review, it would be wrong to exclude potential predictors because earlier there was no statistical significant relationship. 

We did not amend the manuscript on this point.

Editor: 7. Lines 186 to 188: “The extraction and selection will be an iterative process in which searching, selecting, and extracting potential predictors is repeated systematically, to ensure adequate data collection.” It is unclear if such an iterative process makes sense in a review like this one that apparently will not address qualitative data, although this is not clear in the eligibility criteria.

Response: We understand that our description of the study design has created some unclarity. With ‘iterative’ we referred to the fact that we extend the search backward in time in steps until we no longer find new potential predictors (i.e. reach ‘data saturation’). These potential predictors are as such qualitative data. The retrieved potential predictors will be used as input in our machine learning process. As stated above, we expect to find considerable overlap in identified potential predictors in the literature. This method will enable us to efficiently search a broad corpus of literature. As stated above, we are not performing a systematic review. We have clarified this further in the text of the manuscript. See also our response to point 4 above.

To describe our interpretation of data saturation more clearly, we added this at two points in the revised manuscript with track changes (line 49 and 322: “(i.e., no new potential predictors are found)”).

Editor: 8. Lines 188 to 189: “No risk of bias assessment will be performed because no effects on endpoints are quantified.” Systematic reviews must perform evaluations of risk of bias and of the overall certainty of evidence. The authors’ argument that “no effects on endpoints are quantified” as a justification is flawed. The only reason why endpoints are not quantified is that the authors intend only to “count the votes” of included studies in terms of positive, negative, and absent associations. Vote counting is one of the worst possible strategies to perform a literature synthesis and is not acceptable for any high-quality systematic review.

Response: As earlier explained in the answer on points 2 and 7, this review is a scoping review in which we would like to retrieve as many as possible potentially relevant variables to predict the risk of frailty, it is not a systematic review. In a paper of Munn et al. (2018), a table defining the characteristics of different reviews is included. In this table, the authors state that a risk of bias assessment is not mandatory in scoping reviews.

We will indeed count how many times a variable is mentioned and if an association with frailty is present or absent, however this is for administrative reasons only, and NOT to summarize evidence of an association or to draw any conclusions about associations as is done in vote counting. Eventually, this review will result in a set of potentially relevant predictors to use as features in our machine learning process.

We added this in line 234-236 of the revised manuscript with track changes: “The count of how many times a variable is mentioned and if an association with frailty is present or absent will not be used to summarize evidence of an association or to draw any conclusions about associations.”. Furthermore, to emphasize we amended the conclusion of our manuscript in line 327-330 of the revised manuscript with track changes: “The extensive list of potential predictors of frailty provided by this study can be used as an evidence-based foundation for a modeling effort using machine learning to develop algorithms to be used for automated prediction of the risk of frailty based on hospital EHR variables recorded in routine care in the near future.”

Editor: 9. I found reviewer #3 comment on the question of acute care data extremely relevant because it is often important to differentiate frailty present before hospital admission from frailty developed because of an acute problem immediately before or in the course of hospital admission.

In our planned future research, we will build an automated tool to predict the risk of someone being frail. We will use machine learning to develop that automated tool and will include all potential predictors identified through this review while creating the final algorithms. We agree that it is important to differentiate between frailty present before hospital admission or frailty present because of an acute illness. In our future research, we take this concern into account by using labelled data, in which the label is frailty present in steady state.

We added this to line 226-227 of the revised manuscript with track changes: “In the supervised learning methods, frailty will be defined as frailty present in steady state.”.

Editor: 10. I also agree with reviewer #3 that the authors should read Searle’s 2008 article, which provides guidance on the development of frailty indexes, which can also be used with EHR.

Response: We would like to thank the reviewer for suggesting Searle et al. 2008 BMC Geriatrics as a ‘recipe’ for frailty indices. We read it carefully, however, our planned further research is not to develop another frailty index based on deficits. As we intend to use machine learning techniques to build a model that predicts the risk of someone being frail and runs on routine health care data available in the EHR, the crux of such an approach is to use any type of possible predictors to build a model with as high as possible predictive value. The machine learning process will automatically lead to feature selection (i.e., deleting irrelevant variables from the model). Therefore, we will not only use data from frailty indices or frailty measures, but any potential predictor available in the EHR data. Also, we do not focus on a specific category or use any prespecified selection criterion like an association with health status or age.

Response to the reviewers

Reviewers' comments:

Reviewer #1: This is an interesting protocol of a systematic review, in which the authors aim to identify predictors for frailty that could serve as a foundation for future development of EHR-based frailty measures in secondary care. As there has been increasing efforts in various countries to develop automatic frailty tools in clinical settings, this work is important and relevant to the field. I also think that this protocol is well-written, with clear and transparent descriptions on the literature search process, data management, and timeline. I have a few comments below, which I hope can further help in improving the clarity of the aims and methods of the protocol:

Response: First, we would like to thank the reviewer for his/her interest in our review protocol and recognizing its importance and relevance. Second, we would like to thank the reviewer for pointing out the strengths but even more for the comments to help improving our manuscript. As stated before in the response to the editor section, we would like to clarify that our review is a scoping review, not a systematic review. 

We have amended the manuscript to point this out more clearly, most explicitly in line 95-99 of the revised manuscript with track changes: “Therefore, the aim of our study is to perform a scoping review to build the foundation for the development of an automated prediction tool for the risk of frailty on the basis of routine health care data present in the EHR in secondary care, by identifying all potentially relevant features to test as potential predictors in a modeling effort based on machine learning”.

Reviewer #1:- My main concern is on the meaning of the word “predictor”, which is not entirely clear to me in this context of EHR-based frailty. I think it can mean slightly different things depending on the operational approach to frailty. If I understand correctly, the authors would like to identify both risk factors for frailty and variables for creating the frailty measures (lines 85–87)? For the latter, I am not sure if I would call it a predictor if, for example, the health deficit variables in creating a Rockwood frailty index are embedded as part of the frailty construct. I think the authors can perhaps give some more examples of frailty measures and predictors in the Introduction to make it clearer.

We agree with the reviewer that “predictor” can mean different things. Therefore, we would like to explain how we interpret the meaning of the word “predictor” in the context of our machine learning efforts. We consider any variable that is described as potentially related to frailty as a potential predictor for the algorithms we will develop based on EHR data. This will not be restricted to variables used to create frailty measures, health deficits or any other category.

We now describe this more clearly in line 210-213 of the revised manuscript with track changes: “In this study, a variable is a single data point (e.g., question, item, clinical value, or test result) and is considered a potential predictor when it is described as a factor possibly related to frailty, irrespective of the variable type and its described significance in that article.”.

Reviewer #1:- Similarly, I would also expect some descriptions in the Methods on how frailty will be defined. Do the authors plan to include all available frailty measures? “Frailty” can mean quite different things (and the predictors may be different) for a frailty phenotype, a deficit-based frailty index, a data-driven frailty score, or the Clinical Frailty Scale.

Response: We agree with the reviewer that there are different types and measures of frailty, and they can have different predictors. This review is designed to extract all possible predictors of frailty, which for now includes all different types of frailty, using any measure of frailty. 

We added this to line 124-125 of the revised manuscript with track changes: “This is not restricted to any specific type or measure of frailty.”. In our planned further research, those possible predictors will serve as input for prediction modelling using machine learning. It could be that the outcome is that more than one algorithm is needed to yield useful predictions in different clinical situations. We clarified this further in the discussion section of the manuscript and by writing ‘algorithms’ instead of ‘algorithm’ in this response letter and in the manuscript, where applicable.

Reviewer #1:- Another thing I also feel a bit confused is whether the authors are searching only articles for EHR-based frailty measures, or any frailty measures in general that are used in secondary care? I think this can be stated more clearly in the aim.

Response: As mentioned before in the response on point 10 in the editor section, our planned further research is not to develop another frailty index based on deficits. As we intend to use machine learning techniques to build a model that predicts the risk of someone being frail and runs on routine health care data available in the EHR, the crux of such an approach is to use any type of possible predictors to build a model with as high as possible predictive value. The machine learning process will automatically lead to feature selection (i.e., deleting irrelevant variables from the model). Therefore, we will not only use data from frailty indices or frailty measures, but any potential predictor available in the EHR data. We do not limit our search to any type of variable or context. We aim to find all potential predictors for frailty that are described in the literature.

We rephrased the aim to state this more unambiguously in the manuscript (line 95-99 of the revised manuscript with track changes): “Therefore, the aim of our study is to perform a scoping review to build the foundation for the development of an automated prediction tool for the risk of frailty on the basis of routine health care data present in the EHR in secondary care, by identifying all potentially relevant features to test as potential predictors in a modeling effort based on machine learning”.

Reviewer #1:- The authors emphasized a lot on the potential development of an automated frailty detection tool based on the predictors identify from this study. A bit more information on that will be of interest; for example, are the authors referring to some machine learning methods to build automated frailty tools?

Response: Our goal is indeed as the reviewer states. In our further research we aim to build an automated tool for predicting the risk of frailty using machine learning methods. The potential predictors identified in this review will be used as features in the machine learning process. We aim to use different methods of supervised (e.g. regression and random forest) and unsupervised (e.g. classification and clustering) learning.

To clarify this, we added this more extensively and more straightforward to line 224-226 of the revised manuscript with track changes: “In that machine learning process we aim to use different methods of supervised (e.g. regression and random forest) and unsupervised (e.g. classification and clustering) learning.”.

Reviewer #1:- I don’t think it is accurate to say that “none of them [automated tools using EHR data] in secondary care” (line 248). There are at least several EHR-based frailty indices (PMID: 33770164, 34967685, 34650997), as well as the Hospital Frailty Risk Score (PMID: 29706364) developed for frailty detection in secondary care. Please consider rephrasing it.

Response: We agree with the reviewer that there are several EHR-based frailty indices developed in secondary care. We rephrased this sentence and pointed out another issue: in most mentioned references, the method to calculate the frailty score is an unweighted accumulative of deficits, all using a slightly different set of deficits.

This can be found in line 300- 311 of the revised manuscript with track changes: “Many frailty measures have been developed, but most are based on surveys or questionnaires and are labor-intensive to complete. The widespread use of EHRs enables automated prediction of the risk of frailty using routine care data, possibly even without using frailty screening instruments. Some tools using EHR data to predict frailty and with the ability to be fully automated have been proposed, however most of them use unweighted frailty indices like the electronic Frailty Index (eFI), and all with a slightly different set of deficits [9–11,17–19]. Thus, it is not yet clear which included variables using routinely collected data from the hospital EHR will work best. This study provides an overview of all potential predictors of frailty and adds advancements for automated prediction of the risk of frailty in secondary care. This is a promising perspective, being much less labor-intensive compared to screening each individual patient by hand. Also, it may raise awareness of frailty, especially in patients who are not screened for frailty because they seem robust.”. 

Reviewer #1:- The authors may consider stating explicitly in the title that it is a systematic review (and it is frailty for secondary care) to help the readers understand the study design.

Response: We included the design (scoping review, not a systematic review) and the setting (secondary care) in the title.

Reviewer #2: The authors are embarking on a worthwhile endeavor (aimed at better using ROUTINE healthcare data for frailty) but a few points need more careful consideration

Reviewer #2: 1) It is unclear exactly how this proposed review differs from the work of Bery et al (ref 8). Yes the authors want to include more than just frailty instruments - but wouldn't most of the articles yielded by the search terms (Box 1) be frailty scores? If not, then the authors should outline what kind of articles they seek to capture (in addition to frailty scores/measures). I worry that not specifying this will yield a set of articles that is too heterogeneous

Response: We agree with the reviewer that many articles will contain information about frailty scores, however, in our search results we also expect a lot of prospective or retrospective observational research containing potential predictors for frailty. We agree that our search results and the articles we aim to include are more heterogeneous than the articles included in Bery et al. However, we do not see this heterogeneity as a limitation, but as a strength to fully utilize the power of machine learning instead. 

We did not amend the manuscript on this point.

Reviewer #2: 2) More clarification is needed on the endpoint of the search ("data saturation"). The search outlined will possibly yield many thousand articles. What if each article generates new variables not previously seen (note that the "accumulation of deficits frailty approach" gives a near-infinite number of possible deficits). How will they deal with this volume of articles?

Response: As mentioned before in the response on point 10 in the editor section and the answer on point 4 of reviewer #1, our planned further research is not to develop another frailty index based on deficits. However, it is possible that a near-infinite number of possible predictors for frailty can be found. We would like to clarify that if saturation is not reached in a round of the search, we will continue data extraction until data saturation has been reached (or all retrieved articles have been handled). We are aware of the high number of articles that can be found. As we do not have to go through the articles in full detail, but only retrieve all relevant potential predictors, this high number of articles will be feasible to handle.

To describe our interpretation of data saturation more clearly, we added this at two points in the revised manuscript with track changes (line 49 and 322: “(i.e. no new potential predictors are found)”).

Reviewer #2: 3) As the authors may know, there are two major "schools of thought" regarding frailty measurement. The accumulation of deficits approach VS the phenotype approach. In studies such as the present study, the accumulation of deficits approach will be OVER-represented as these frailty scores can use a wide variety of variables (in fact, any variable is allowed as long as there are enough deficits). The authors should outline whether they will record the approach used for each study their review identifies.

Response: We agree with the reviewer that in our search particularly many deficits will arise. As earlier explained in our comments to reviewer 1, it is our aim to give a complete overview of all potential relevant predictors, and we do not limit our search to any type of variable (see also our answer to point 1 from Reviewer #2). Therefore, we will not record which approach was used for which study included in our review, as this is not relevant for our scoping review and research. 

Reviewer #2: 4) Risk of bias - the authors should consider doing a more systematic "risk of bias" assessment (rather than just discrepancy checking amongst the authors). It will be helpful to consider how to grade the quality of articles included in the final review

Response: For the answer to this point we would like to refer to point 8 in the response to the editor section. This review is a scoping review in which we would like to retrieve as many as possible potentially relevant variables to predict the risk of frailty. A risk of bias assessment is not mandatory in a scoping review.

Reviewer #3: This manuscript summarizes the protocol for a literature review to identify predictors as part of automated frailty detection using the electronic health record. Authors attest that results reported have not been published elsewhere.

Reviewer #3: The process described will include a systematic review and will consider multiple possible sources of data from the inpatient hospital sphere for frailty.

Response: This review is intended to be a scoping review, not a systematic review, in which we would like to retrieve as many as possible potentially relevant variables to predict the risk of frailty. This review will result in a set of potentially relevant variables which can be used as potential predictors in our planned machine learning efforts, however these variables are not only derived from the inpatient hospital sphere as the reviewer states.

Reviewer #3: Major Concerns:

Reviewer #3: 1. While frailty is defined correctly according to the most prominent definitions, the manuscript authors do not specifically lay out how their predictors will be selected beyond that they have been used elsewhere. This is unfortunate, because the concept of frailty detection using the frailty index methodology is well described and - significantly - shows that it is vitally important that the elements selected are related to aging. Strongly recommend that the authors review Searle et al. 2008 BMC Geriatrics for a "recipe" for a frailty index.

Response: We would like to thank the reviewer for sharing his/her concerns regarding our review protocol. We appreciate the recommendation of Searle et al., however it is not our aim to build a frailty index, but to build a fully automatic tool to predict the risk of frailty based on EHR data using machine-learning modeling. This is a very different approach; we found the article of Searle et al. helpful, yet not relevant for our research plan. In our review, we therefore do not restrict our search by looking for age-related variables only. Furthermore, using machine learning techniques can sometimes bring up unexpected predictors. Restricting to age-related variables only could probably also prevent gaining surprising insights.

Reviewer #3: 2. The authors note that most frailty assessments and indices are in community care (primary care in US). They note two particular frailty indices used in UK (Clegg et al) and US (Pajewski et al). Both use routinely collected data. However, they do not engage with the question of using acute care data (when the patient is not in a "steady state" for frailty assessment. Does this truly reflect frailty, or does it reflect acuity of illness? some consideration of this concern should be integrated into the plan.

Response: For our response to this question, we would like to refer to point 9 in the response to the editor section. In our planned future research, we will build an automated tool to predict the risk of someone being frail. We will use machine learning to develop that automated tool and will include all potential predictors identified through this review while creating the final algorithms. We agree that it is important to differentiate between frailty present before hospital admission or frailty present because of an acute illness. In our future research, we take this concern into account by using labelled data, in which the label is frailty present in steady state.

We added this to line 226-227 of the revised manuscript with track changes: “In the supervised learning methods, frailty will be defined as frailty present in steady state.”.

Reviewer #3: 3. No description is listed for how authors will determine whether or not to include elements in their frailty assessment, or how closely related a concept is to frailty, or whether to weight items as in traditional risk prediction models or weight all as "1" as in frailty indices

Response: As mentioned in the answer to point 1 of reviewer #3, it is not our intention to build a new frailty index. Moreover, we aim to build a prediction model using machine learning techniques. As earlier explained, the predictors included in the final algorithms will be selected by using different types of learning methods. Weighting and feature selection is part of a machine learning process.

Reviewer #3: 4. The article is clear and is written in standard English.

Response: We would like to thank the reviewer for this comment, which enables us to focus on the contents.

Reviewer #3: Given the above serious concerns, I would not recommend this for publication until and unless the authors engage with the extant literature on hospital-based frailty indices, the pros/cons of defining frailty using acute hospital (versus primary care) data, and the existing guidelines for building a frailty index. 

Response: We hope to have taken away the reviewer’s concerns by providing our rebuttal to each point in this letter and the revisions we made in our manuscript. Specifically, we stated the aim more unambiguously, explained our rationale, and improved the methods. We would like to genuinely thank the editor and the reviewers for concerning these revisions.

Sincerely,

Dirk H. van Dalen, RN, MSc

Prof. dr. Esther de Vries MD PhD

Jeroen Bosch Academy Research, Jeroen Bosch Hospital, ‘s-Hertogenbosch & Tranzo, Tilburg School of Social and Behavioral Sciences, Tilburg University, Tilburg, The Netherlands

PO Box 90153, 5200ME ‘s-Hertogenbosch, The Netherlands

Phone 0031735533114

Email: e.d.vries@jbz.nl

---

## [Decision Letter · Decision Letter 1]

15 Aug 2022

PONE-D-21-38902R1Registered report protocol: a scoping review to identify potential predictors as features for automated prediction of the risk of frailty in secondary carePLOS ONE

Dear Dr. de Vries,

Thank you for submitting your manuscript to PLOS ONE. After careful consideration, we feel that it has merit but does not fully meet PLOS ONE’s publication criteria as it currently stands. Therefore, we invite you to submit a revised version of the manuscript that addresses the points raised during the review process.

Please see the end of this message for specific editorial comments.

We look forward to receiving your revised manuscript.

Kind regards,

Edison I.O. Vidal, MD, MPH, PhD

Section Editor

PLOS ONE

Additional Editor Comments:

I have assessed the reviewers’ comments and revised the new version of the manuscript myself. Unfortunately, there are still some  major methodological concerns that must be addressed before the manuscript can be accepted as a registered protocol.

1. I agree with reviewer #3 when they write that it is still unclear whether the authors wish to identify predictors of being frail or of becoming frail. This is a critical issue that was already present in the first round of reviews, when reviewers asked the authors to clarify whether they aimed to detect frail patients or to detect patients at risk of developing frailty. That lack of clarity is present in the authors’ response to my first comment, when they wrote that they “intend to carry out this review to gather input for building an automated tool to predict the risk of someone being frail, not to develop another frailty detection instrument”. Unfortunately, “a frailty detection instrument” can be defined as a tool to predict whether someone is frail. Hence, the authors decision to replace “frailty detection tool” with “tool for prediction of the risk of frailty” did not solve the problem raised in the first round of review.

The authors now stated that “the aim of our study is to perform a scoping review to build the foundation for the development of an automated prediction tool for the risk of frailty on the basis of routine health care data present in EHR in secondary care, by identifying all potentially relevant features to test as potential predictors in a modelling effort based on machine learning.”

To “predict the risk of frailty” is too vague because it can mean both identifying people who are frail at this moment, as well as to predict who will develop frailty within a certain timeframe. Those remain two different goals that require different methods and justifications. For example, in the end of the introduction section the authors argued that their review will differ from two previous reviews on this subject because they “extracted variables from frailty instruments” whereas the authors aim to “identify all potentially relevant predictors for frailty that can be extracted from the hospital EHR, not limited to frailty instruments only”. Taking into account that most Frailty Indices were constructed based on EHR, it is likely that the results of this review will also include many variables already used to construct those indices and that most of their results will derive from those studies. Hence, unless the authors’ aim to predict the risk of people becoming frail in the future, it is unclear why the results of the two previous reviews cannot be used to feed their future study using machine learning.

The authors should also recognize that there are some very simple frailty detection tools available, such as the Study of Osteoporotic Fractures and the PRISMA-7 tools, which can be applied in less than four minutes by any healthcare professional.

2. Reviewers have requested the authors to specify which kind of frailty definition they intend to adopt as a gold standard. For the purposes of the scoping review the authors intend to conduct, I understand that they may include studies that used different definitions of frailty. However, it is essential that they add a description to their methods explaining that when summarising the literature, they will present the prediction variables according to the definition of frailty used in the original studies they were extracted from. I agree with the reviewers that, in the next phase of their research, when they will use artificial intelligence to develop a model to predict the presence of frailty, they will have to define upfront what kind of definition of frailty they will use as the outcome of their prediction models. However, I understand that the authors may choose not to make that decision at the present moment. The downside is that their review will probably be more difficult to perform than if they decided to restrict their inclusion criteria to predictors of a specific type of frailty.

3. While reading the authors responses, it appears that there are some misconceptions about the differences between systematic reviews and scoping reviews. Scoping reviews usually have a broader focus than systematic reviews. For example, both systematic reviews and scoping reviews should strive to perform exhaustive searches of the literature. The data saturation approach proposed by the reviewers is not compatible with the methodological expectations of both kinds of reviews. The argument described by the authors stating that “Moreover, by not continuing the search after data saturation has been reached, we prevent an enormous amount of work that would not lead to any increase in the yield of our research” is not compatible with PLOS ONE’s expectations of high methodological standards. I also find that the justification not to search the Web of Science database because the authors are conducting a scoping review inappropriate.

4. The “synthesis of results” section does not provide sufficient information regarding how the author intend to report their results. “The results will be described in an extensive categorized table of all unique potential predictors for frailty, including total count of articles in which a potential predictor was

mentioned and other relevant information.” Specifically, what kind of “other relevant information” do the authors can anticipate to include in those tables? Will they take into account what kind of reference standard was used to define frailty? Will they describe the population of patients that was used in each study?

5. Lines 104 to 105: “Eligible articles are written in English or Dutch and should have an available full text to enable complete review.” That sentence remains inappropriate even after the authors response. If a reference cannot be retrieved by the Dutch Royal Library or from the authors, there is still no reason to say that it was not eligible for the review. It would be more accurate to simply explain that it was not retrieved, as expected by the recommendations of the PRISMA 2020 statement.

Reviewers' comments:

Reviewer's Responses to Questions

**Comments to the Author**

1. Does the manuscript provide a valid rationale for the proposed study, with clearly identified and justified research questions?

Reviewer #1: Yes

Reviewer #2: Partly

Reviewer #3: Partly

2. Is the protocol technically sound and planned in a manner that will lead to a meaningful outcome and allow testing the stated hypotheses?

Reviewer #1: Yes

Reviewer #2: Partly

Reviewer #3: Partly

3. Is the methodology feasible and described in sufficient detail to allow the work to be replicable?

Reviewer #1: Yes

Reviewer #2: No

Reviewer #3: Yes

4. Have the authors described where all data underlying the findings will be made available when the study is complete?

Reviewer #1: Yes

Reviewer #2: No

Reviewer #3: Yes

5. Is the manuscript presented in an intelligible fashion and written in standard English?

Reviewer #1: Yes

Reviewer #2: Yes

Reviewer #3: Yes

6. Review Comments to the Author

You may also provide optional suggestions and comments to authors that they might find helpful in planning their study.

Reviewer #1: I would like to thank the authors for the detailed response and clarifications. The manuscript has much improved and the remaining issues are out of the scope of this paper.

One final comment: while I understand that the aim of the study is to identify predictors of frailty in EHR data, it is still not entirely clear to me how “frailty” is defined in this context (e.g., whether it is considered as a deficit-accumulated frailty index, phenotypic frailty, or clinical frailty). Especially when using machine learning methods in the authors’ future work, which outcome/measure will be used as the reference standard to define frailty? This may perhaps be less relevant to the current protocol for the scoping review, but I encourage the authors to think about it and define more clearly on what they mean by “frailty” in their work.

Reviewer #2: Based on the changes to the manuscript, I now better understand the rationale for this work, which is to identify "predictors of frailty" which will then be fed into a machine learning algorithm and used on real patient data.

However, I still have some concerns about the proposed methodology.

1) "Data saturation"  having had personal experience with extracting frailty predictors from the literature, the volume of articles is incredible, and increases each year. This is especially true of articles deriving FIs based on the accumulation of deficits approach. I do not think saturation will be reached by the authors' current definition. There will always be one more FI that has a few more variables. I suggest the authors revise their plan to end the search

2) Definition of types of articles that will be found in the search  the authors should better define the kinds of articles they are hoping their search will find. Right now it is a bit vague. It is not just frailty measures, but quite a heterogeneous groups of articles on older people and assessment. In plain English under "literature search" they should spell this out

Reviewer #3: 1) What led your team to choose a scoping review over a systematic review? The field of frailty is not new, and it seems that the team is ready to do a systematic review to inform research, rather than a scoping review to describe the state of the field...

2) If you are developing a prediction tool for frailty, rather than a frailty index, then you will need to choose your "gold standard" definition of frailty. How do you intend to do so? using which definition? how do you separate out the elements that define frailty (deficits in n index, features in FRAIL scale or Fried's phenotype) vs. upstream predictors of future frailty?

3) Again, in reviewing response to questions - are you differentiating between someone BEING frail vs. BECOMING frail at a point in the future? How are you defining whether frailty is "present in steady state"?

7. PLOS authors have the option to publish the peer review history of their article (what does this mean?). If published, this will include your full peer review and any attached files.

Reviewer #1: **Yes: **Jonathan Mak

Reviewer #2: No

Reviewer #3: No

---

## [Author Response · Author response to Decision Letter 1]

2 Sep 2022

Dear editor, dear reviewers,

Thank you again for the useful comments. Hereby we submit our second revision of the manuscript. Our revisions can be found in the tracked changes in the manuscript. Our response to the comments and suggestions made by the editor and reviewers can be found below, in italics. Thank you for your consideration of these responses and revisions.

Response to the editor

Additional Editor Comments:

I have assessed the reviewers’ comments and revised the new version of the manuscript myself. Unfortunately, there are still some major methodological concerns that must be addressed before the manuscript can be accepted as a registered protocol.

Editor 1a. I agree with reviewer #3 when they write that it is still unclear whether the authors wish to identify predictors of being frail or of becoming frail. This is a critical issue that was already present in the first round of reviews, when reviewers asked the authors to clarify whether they aimed to detect frail patients or to detect patients at risk of developing frailty. That lack of clarity is present in the authors’ response to my first comment, when they wrote that they “intend to carry out this review to gather input for building an automated tool to predict the risk of someone being frail, not to develop another frailty detection instrument”. Unfortunately, “a frailty detection instrument” can be defined as a tool to predict whether someone is frail. Hence, the authors decision to replace “frailty detection tool” with “tool for prediction of the risk of frailty” did not solve the problem raised in the first round of review.

Response: We would like to clarify that it is our intention to predict the risk of someone being frail, not the risk of someone becoming frail. We do this with the objective to support healthcare professionals in making choices in diagnosing and treating patients’ health problems while considering the possible presence of frailty. The outcome of our future algorithm will be an estimation of the probability that someone is frail. Above a certain cut-off in this probability estimation (to be determined in a future study in collaboration with the domain experts), these patients can then be referred to the department of geriatrics for detecting whether frailty is indeed present, and for subsequent frailty management.

To clarify this we made some changes at several points in the manuscript, most explicitly in the title: “Registered report protocol: a scoping review to identify potential predictors as features for developing automated estimation of the probability of being frail in secondary care.”; in line 33 of the revised manuscript with track changes: “We aim to identify potential predictors that could be used as features for modeling algorithms on the basis of routine hospital EHR data to incorporate in an automated tool for estimating the probability of being frail.”; in line 89 of the revised manuscript with track changes: “Therefore, the aim of our study is to perform a scoping review to build the foundation for the development of an automated tool for estimating the probability of being frail on the basis of routine health care data present in the EHR in secondary care, by identifying all potentially relevant features to test as potential predictors in a modeling effort based on machine learning.”; and in line 318 of the revised manuscript with track changes: “The extensive list of potential predictors of frailty provided by this study can be used as an evidence-based foundation for a modeling effort using machine learning to develop algorithms to be used for automated estimation of the probability of being frail based on hospital EHR variables recorded in routine care in the near future.”

Editor 1b. The authors now stated that “the aim of our study is to perform a scoping review to build the foundation for the development of an automated prediction tool for the risk of frailty on the basis of routine health care data present in EHR in secondary care, by identifying all potentially relevant features to test as potential predictors in a modelling effort based on machine learning.”

To “predict the risk of frailty” is too vague because it can mean both identifying people who are frail at this moment, as well as to predict who will develop frailty within a certain timeframe. Those remain two different goals that require different methods and justifications. For example, in the end of the introduction section the authors argued that their review will differ from two previous reviews on this subject because they “extracted variables from frailty instruments” whereas the authors aim to “identify all potentially relevant predictors for frailty that can be extracted from the hospital EHR, not limited to frailty instruments only”. Considering that most Frailty Indices were constructed based on EHR, it is likely that the results of this review will also include many variables already used to construct those indices and that most of their results will derive from those studies. Hence, unless the authors’ aim to predict the risk of people becoming frail in the future, it is unclear why the results of the two previous reviews cannot be used to feed their future study using machine learning.

Response: As mentioned in the response on comment 1a of the Editor, it is our intention to estimate the probability of someone being frail, not the risk of someone becoming frail. We do this with the objective to support healthcare professionals in making choices in diagnosing and treating patients’ health problems while considering the possible presence of frailty. The outcome of our future algorithm will be an estimation of the probability that someone is frail. Above a certain cut-off in this probability estimation (to be determined in a future study in collaboration with the domain experts), these patients can then be referred to the department of geriatrics for detecting whether frailty is indeed present, and for subsequent frailty management.

We agree that it is likely that the results of our review will also include many variables derived from studies into the construction of frailty indices. However, with our search strategy we also hope to derive a considerable number of variables from other types of research which will complement our search for potential predictors to add as features in our machine learning efforts. Both mentioned review articles (Bouillon et al. 2013, searched since inception until 2011; Bery et al. 2020, searched since 2011 until March 2018) (and the articles included therein) will now be the starting point of our search, consistent with our response to the further points (editor point 3; reviewer #2, point 1) regarding the search strategy and data saturation approach, in response to which we changed our search strategy.

We changed our manuscript on this point, most explicitly in line 41 of the revised manuscript with track changes: “Second, we add two published literature reviews (and the articles included therein) [Bery 2020; Bouillon, 2013] to our search results. In these reviews, articles on potential predictor variables in frailty screening tools were included from inception until March 2018.”; in line 100 of the revised manuscript with track changes: “Therefore, we will perform an extensive search in a broad corpus of published literature since 2018. The earlier mentioned reviews (and the articles included therein) will then be added to our search results [8,13]. We strive to identify all potentially relevant predictors for the presence of frailty that can be extracted from the hospital EHR, not limited to frailty instruments only.”; in line 154 of the revised manuscript with track changes: “The review articles of Bery (2020) and Bouillon (2013) (and the articles included therein) will be added to our search results [8,13]. The goal is to identify and extract all potential predictors of the presence of frailty.”; and by deleting lines 164 from the revised manuscript with track changes: “Searching is continued until data saturation is reached. Data saturation is considered to have been reached when no new possible predictors can be found in the included primary research articles anymore. After the analysis of primary research articles from the first run (published ≥2018), the review articles from that run will be analyzed, starting with the most recent one. As review articles include earlier published literature, the search will be further extended if new possible predictors are found in the review articles, adding one year to the PubMed (MEDLINE), CINAHL Plus, and Embase search strategies in each new run until data saturation has been reached (i.e. the second run searching articles published in 2017 and check for data saturation, the third run searching articles published in 2016 and check for data saturation, and so on, including the process with review articles from the same search period each time). If no new possible predictors are found in the review articles in a run, data saturation is considered to be confirmed, and the search will close.”

Furthermore, we amended and updated figure 2 and the appendices 2 and 3 containing the search strategies. 

Editor 1c. The authors should also recognize that there are some very simple frailty detection tools available, such as the Study of Osteoporotic Fractures and the PRISMA-7 tools, which can be applied in less than four minutes by any healthcare professional.

Response: We do recognize that there are several very simple frailty detection tools available, however, even though such a tool can be applied in less than four minutes, the amount of time is still considerable as it is nearly half of the total time a doctor has for a regular outpatient visit. Therefore, we believe an algorithm that gives an estimation of the probability of the presence of frailty in a patient is relevant, by increasing the number of frail patients that receive care adjusted to their needs.

 We did not change our manuscript on this point.

Editor 2. Reviewers have requested the authors to specify which kind of frailty definition they intend to adopt as a gold standard. For the purposes of the scoping review the authors intend to conduct, I understand that they may include studies that used different definitions of frailty. However, it is essential that they add a description to their methods explaining that when summarizing the literature, they will present the prediction variables according to the definition of frailty used in the original studies they were extracted from. I agree with the reviewers that, in the next phase of their research, when they will use artificial intelligence to develop a model to predict the presence of frailty, they will have to define upfront what kind of definition of frailty they will use as the outcome of their prediction models. However, I understand that the authors may choose not to make that decision at the present moment. The downside is that their review will probably be more difficult to perform than if they decided to restrict their inclusion criteria to predictors of a specific type of frailty.

Response: We agree with the editor that not choosing a specific type of frailty for this review may result in a more difficult performance of the review, however, we do believe it will strengthen our future research if we currently make no restrictions for retrieving potential variables for our machine learning effort. Furthermore, using artificial intelligence in a large sample of patients will allow us to use variables retrieved from all types of frailty measurements at the same time as well as combine these with other variables retrieved from the experience of domain experts. In the supervised machine learning process, we will use the reported results of available frailty screening tool(s) in the patient’s EHR as outcome variable(s). Finally, we intend to compare the outcome of our developed algorithm (the estimated probability of the presence of frailty) with frailty measured by comprehensive geriatric assessment (CGA) (if available in the patient’s EHR) and/or consensus among (inter)national domain experts upon reviewing the patient’s EHR as ‘gold standard’. 

We added the definition of frailty used in an article as variable to extract in table 1; in line 200 of the revised manuscript with track changes: “The metadata of the articles (i.e., author(s), year of publication, study name, study design, study setting, and study country), descriptive baseline variables (i.e., study population, age, sex, and number of subjects), and all unique potential predictors of being frail (including relevant information such as the potential predictor type and the definition of frailty used in the article a potential predictor was mentioned; Table 1) will be extracted and recorded in Microsoft Excel.”; and in line 222 of the revised manuscript with track changes: “The results will be described in an extensive categorized table of all unique potential predictors for being frail, including total count of articles in which a potential predictor was mentioned and other relevant information such as the potential predictor type and the definition of frailty used in the article where a potential predictor was mentioned.”.

Editor 3. While reading the authors responses, it appears that there are some misconceptions about the differences between systematic reviews and scoping reviews. Scoping reviews usually have a broader focus than systematic reviews. For example, both systematic reviews and scoping reviews should strive to perform exhaustive searches of the literature. The data saturation approach proposed by the reviewers is not compatible with the methodological expectations of both kinds of reviews. The argument described by the authors stating that “Moreover, by not continuing the search after data saturation has been reached, we prevent an enormous amount of work that would not lead to any increase in the yield of our research” is not compatible with PLOS ONE’s expectations of high methodological standards. I also find that the justification not to search the Web of Science database because the authors are conducting a scoping review inappropriate.

Response: As mentioned in the response on editor 1b, we understand that the data saturation approach does not meet PLOS ONE’s expectations of high methodological standards. Also considering point 1 of reviewer #2 on the achievability of this approach, we have changed our search strategy. We let go the data saturation approach. Considering the current point raised by the editor, both review articles mentioned in 1b (Bouillon et al. 2013, searched since inception until 2011; Bery et al. 2020, searched since 2011 until March 2018) (and the articles included therein) will now be the starting point of our search. From that point on (2018 until present time), we will perform our own literature search. We also included searching Web of Science in our new search strategy.

We changed the methods section of our manuscript on this point, most explicitly in line 118 of the revised manuscript with track changes: “Four databases will be searched for eligible articles: PubMed (MEDLINE), CINAHL Plus, Embase, and Web of Science.”; in line 154 of the revised manuscript with track changes: “The review articles of Bery (2020) and Bouillon (2013) (and the articles included therein) will be added to our search results.”; in line 255 of the revised manuscript with track changes: “An exploratory search in PubMed (MEDLINE), CINAHL Plus, Embase, and Web of Science resulted in a total of 32.526 records (7.796 PubMed (MEDLINE), 2.722 CINAHL Plus, 6.147 Embase, and 15.861 Web of Science, these were not yet checked for duplicate records) (Fig 2). ”; and by removing the first limitation in line 306 of the revised manuscript with track changes: “First, the literature is not searched across all available databases, which could result in missing articles and therefore potential predictors. In our opinion, the impact of this limitation is small, as we expect considerable overlap in the described potential predictors between the published articles”.

Furthermore, we amended and updated figure 2 and the appendices 2 and 3 containing the search strategies.

Editor 4. The “synthesis of results” section does not provide sufficient information regarding how the author intend to report their results. “The results will be described in an extensive categorized table of all unique potential predictors for frailty, including total count of articles in which a potential predictor was

mentioned and other relevant information.” Specifically, what kind of “other relevant information” do the authors can anticipate to include in those tables? Will they take into account what kind of reference standard was used to define frailty? Will they describe the population of patients that was used in each study?

Response: We agree with the editor that ‘other relevant information’ is a bit vague. The reason we kept this open is that we do not want to limit the way we report our synthesis beforehand. Furthermore, the essence of the machine learning approach in our future research is to use as many potential predictor variables as possible as features. The variables with the least predictive value will then automatically be dropped in the feature selection process.

As earlier mentioned in the response to comment 2 of the Editor, we added the definition of frailty used in an article as variable to extract in the methods section in line 200 of the revised manuscript with track changes: “The metadata of the articles (i.e., author(s), year of publication, study name, study design, study setting, and study country), descriptive baseline variables (i.e., study population, age, sex, and number of subjects), and all unique potential predictors of being frail (including relevant information such as the potential predictor type and the definition of frailty used in the article a potential predictor was mentioned; Table 1) will be extracted and recorded in Microsoft Excel.”; and in line 222 of the revised manuscript with track changes: “The results will be described in an extensive categorized table of all unique potential predictors of being frail, including total count of articles in which a potential predictor was mentioned and other relevant information such as the potential predictor type and the definition of frailty used in the article where a potential predictor was mentioned.”

Editor 5. Lines 104 to 105: “Eligible articles are written in English or Dutch and should have an available full text to enable complete review.” That sentence remains inappropriate even after the authors response. If a reference cannot be retrieved by the Dutch Royal Library or from the authors, there is still no reason to say that it was not eligible for the review. It would be more accurate to simply explain that it was not retrieved, as expected by the recommendations of the PRISMA 2020 statement.

Response: We agree with the editor that it would be more accurate to mention articles as not retrieved instead of not eligible in case full texts cannot be retrieved.

 We changed our manuscript on this point in line 120 of the revised manuscript with track changes: “Articles that we were not able to get hold of as full-text will be reported as not retrieved.”

Reviewer #1: I would like to thank the authors for the detailed response and clarifications. The manuscript has much improved and the remaining issues are out of the scope of this paper.

Response: We would like to thank reviewer #1 for his positive reaction on our previous response.

1) One final comment: while I understand that the aim of the study is to identify predictors of frailty in EHR data, it is still not entirely clear to me how “frailty” is defined in this context (e.g., whether it is considered as a deficit-accumulated frailty index, phenotypic frailty, or clinical frailty). Especially when using machine learning methods in the authors’ future work, which outcome/measure will be used as the reference standard to define frailty? This may perhaps be less relevant to the current protocol for the scoping review, but I encourage the authors to think about it and define more clearly on what they mean by “frailty” in their work.

Response: In the supervised machine learning process, we will use the available reported results of frailty screening tool(s) in the patient’s EHR as outcome variable(s). Finally, we intend to compare the outcome of our developed algorithm (the estimated probability of the presence frailty) with frailty measured by comprehensive geriatric assessment (CGA) (if available in the patient’s EHR) and/or consensus among (inter)national domain experts upon reviewing the patient’s EHR as ‘gold standard’.

We added this to line 212 of the revised manuscript with track changes: “In the supervised machine learning methods, frailty will be defined as frailty present in steady state measured by the available reported results of frailty screening tools in the patient’s EHR. Finally, we intend to compare the outcome of the developed algorithm with frailty measured by comprehensive geriatric assessment (CGA) (if available in the patient’s EHR) and/or consensus among (inter)national domain experts upon reviewing the patient’s EHR as ‘gold standard.”.

Reviewer #2:

Based on the changes to the manuscript, I now better understand the rationale for this work, which is to identify "predictors of frailty" which will then be fed into a machine learning algorithm and used on real patient data.

Response: We would like to thank reviewer #2 for his/her additional comments and questions and for acknowledging that the rationale of our manuscript is now more understandable.

However, I still have some concerns about the proposed methodology.

1) "Data saturation"  having had personal experience with extracting frailty predictors from the literature, the volume of articles is incredible, and increases each year. This is especially true of articles deriving FIs based on the accumulation of deficits approach. I do not think saturation will be reached by the authors' current definition. There will always be one more FI that has a few more variables. I suggest the authors revise their plan to end the search

Response: As mentioned in our response to editor’s points 1b and 3, we have changed our search strategy. We let go the data saturation approach. Both review articles mentioned in the response to comment 1b of the editor (Bouillon et al. 2013, searched since inception until 2011; Bery et al. 2020, searched since 2011 until March 2018) (and the articles included therein) will now be the starting point of our search. From that point on (March 2018 until present time), we will perform our own literature search.

We changed our manuscript on this point, most explicitly in line 154 of the revised manuscript with track changes: “The review articles of Bery (2020) and Bouillon (2013) (and the articles included therein) will be added to our search results [8,13].”; and by deleting lines 164 of the revised manuscript with track changes: “Searching is continued until data saturation is reached. Data saturation is considered to have been reached when no new possible predictors can be found in the included primary research articles anymore. After the analysis of primary research articles from the first run (published �2018), the review articles from that run will be analyzed, starting with the most recent one. As review articles include earlier published literature, the search will be further extended if new possible predictors are found in the review articles, adding one year to the PubMed (MEDLINE), CINAHL Plus, and Embase search strategies in each new run until data saturation has been reached (i.e. the second run searching articles published in 2017 and check for data saturation, the third run searching articles published in 2016 and check for data saturation, and so on, including the process with review articles from the same search period each time). If no new possible predictors are found in the review articles in a run, data saturation is considered to be confirmed, and the search will close.”

Furthermore, we amended and updated figure 2 and the appendices 2 and 3 containing the search strategies. 

2) Definition of types of articles that will be found in the search  the authors should better define the kinds of articles they are hoping their search will find. Right now it is a bit vague. It is not just frailty measures, but quite a heterogeneous groups of articles on older people and assessment. In plain English under "literature search" they should spell this out

Response: As shown in figure 1, we did explain the selection criteria in detail. As mentioned before in our response to comments 1b and 2 of the Editor, it is indeed likely that the results of our review will also include many variables derived from studies into the construction of frailty indices. However, with our search strategy we also want to derive a considerable number of variables from other types of research which will complement our search for potential predictors to add as features in our machine learning efforts. Making no restrictions for retrieving potential variables (based on a specific type of frailty or frailty measurement) will strengthen our future research. The articles we look for are all those about elderly patients in which frailty is described as dependent variable and in which the independent variables for frailty are described. We expect most results will be quantitative, observational studies. We do not want to include case studies.

We did not change our manuscript on this point.

Reviewer #3:

1) What led your team to choose a scoping review over a systematic review? The field of frailty is not new, and it seems that the team is ready to do a systematic review to inform research, rather than a scoping review to describe the state of the field...

Response: Indeed, the field of frailty is not new. However, Munn et al. (Systematic review or scoping review? Guidance for authors when choosing between a systematic or scoping review approach (2018), doi: https://doi.org/10.1186/s12874-018-0611-x) state that one of the indications for performing a scoping review is to identify key characteristics or factors related to a concept. As it is our aim to identify as many factors related to the concept of (the presence of) frailty as possible – to be used as features in our machine learning efforts – we do believe that a scoping review approach is valid. 

We did not change our manuscript on this point.

2) If you are developing a prediction tool for frailty, rather than a frailty index, then you will need to choose your "gold standard" definition of frailty. How do you intend to do so? using which definition? how do you separate out the elements that define frailty (deficits in n index, features in FRAIL scale or Fried's phenotype) vs. upstream predictors of future frailty?

Response: As mentioned in our responses to editor’s point 2 and reviewer #1, we will use the available reported results of frailty screening tool(s) in the patient’s EHR as outcome variable(s) in the supervised machine learning process. Finally, we intend to compare the outcome of our developed algorithm (the estimated probability of frailty) with frailty measured by comprehensive geriatric assessment (CGA) (if available in the patient’s EHR) and/or consensus among (inter)national domain experts upon reviewing the patient’s EHR as ‘gold standard’. 

We added this to line 212 of the revised manuscript with track changes: “In the supervised machine learning methods, frailty will be defined as frailty present in steady state measured by the available reported results of frailty screening tools in the patient’s EHR. Finally, we intend to compare the outcome of the developed algorithm with frailty measured by comprehensive geriatric assessment (CGA) (if available in the patient’s EHR) and/or consensus among (inter)national domain experts upon reviewing the patient’s EHR as ‘gold standard”.

3) Again, in reviewing response to questions - are you differentiating between someone BEING frail vs. BECOMING frail at a point in the future? How are you defining whether frailty is "present in steady state"?

Response: As mentioned in our response to editor’s point 1a, we would like to clarify that it is our intention to estimate the probability of someone being frail, not of someone becoming frail. We do this with the objective to support healthcare professionals in making choices in diagnosing and treating patients’ health problems while considering the possible presence of frailty. The outcome of our future algorithm will be an estimation of the probability that someone is frail. Above a certain cut-off in this probability estimation (to be determined in a future study in collaboration with the domain experts), these patients can then be referred to the department of geriatrics for detecting whether frailty is indeed present, and for subsequent frailty management.

To clarify this we made some changes at several points in the manuscript, most explicitly in the title: “Registered report protocol: a scoping review to identify potential predictors as features for developing automated estimation of the probability of being frail in secondary care.”; in line 33 of the revised manuscript with track changes: “We aim to identify potential predictors that could be used as features for modeling algorithms on the basis of routine hospital EHR data to incorporate in an automated tool for estimating the probability of being frail.”; in line 89 of the revised manuscript with track changes: “Therefore, the aim of our study is to perform a scoping review to build the foundation for the development of an automated tool for estimating the probability of being frail on the basis of routine health care data present in the EHR in secondary care, by identifying all potentially relevant features to test as potential predictors in a modeling effort based on machine learning.”; and in line 318 of the revised manuscript with track changes: “The extensive list of potential predictors of frailty provided by this study can be used as an evidence-based foundation for a modeling effort using machine learning to develop algorithms to be used for automated estimation of the probability of being frail based on hospital EHR variables recorded in routine care in the near future.”

---

## [Editor Report · Decision Letter 2]

14 Sep 2022

Registered report protocol: a scoping review to identify potential predictors as features for developing automated estimation of the probability of being frail in secondary care.

PONE-D-21-38902R2

Dear Dr. de Vries,

We’re pleased to inform you that your manuscript has been judged scientifically suitable for publication and will be formally accepted for publication once it meets all outstanding technical requirements.

Kind regards,

Edison I.O. Vidal, MD, MPH, PhD

Section Editor

PLOS ONE
---

## [Editor Report · Acceptance letter]

19 Sep 2022

PONE-D-21-38902R2 

Registered report protocol: a scoping review to identify potential predictors as features for developing automated estimation of the probability of being frail in secondary care. 

Dear Dr. de Vries:

I'm pleased to inform you that your manuscript has been deemed suitable for publication in PLOS ONE. Congratulations! Your manuscript is now with our production department. 

Kind regards, 

on behalf of

Professor Edison I.O. Vidal 

Section Editor

PLOS ONE